

# Alterations of the gut microbiota in type 2 diabetics with or without subclinical hypothyroidism

Yanrong Lv[1,*], Rong Liu[1,*], Huaijie Jia[2], Xiaolan Sun[1], Yuhan Gong[1], Li Ma[1], Wei Qiu[3] and Xiaoxia Wang[1]

[1] School of Public Health, Lanzhou University, Lanzhou, China
[2] State Key Laboratory of Veterinary of Etiological Biology, Lanzhou Veterinary Research Institute, Chinese Academy of Agricultural Sciences, Lanzhou, China
[3] Department of Endocrinology, Xinxiang First People's Hospital, The Affiliated People's Hospital of Xinxiang Medical University, Xinxiang, China
[*] These authors contributed equally to this work.

Corresponding authors
Wei Qiu, qiuwei_endocrine@163.com
Xiaoxia Wang,
wangxiaoxia@lzu.edu.cn

## ABSTRACT

**Background**. Diabetes and thyroid dysfunction are two closely related endocrine diseases. Increasing evidences show that gut microbiota plays an important role in both glucose metabolism and thyroid homeostasis. Meanwhile, copy number variation (CNV) of host salivary $\alpha$-amylase gene (AMY1) has been shown to correlate with glucose homeostasis. Hence, we aim to characterize the gut microbiota and CNV of AMY1 in type 2 diabetes (T2D) patients with or without subclinical hypothyroidism (SCH).

**Methods**. High-throughput sequencing was used to analyze the gut microbiota of euthyroid T2D patients, T2D patients with SCH and healthy controls. Highly sensitive droplet digital PCR was used to measure AMY1 CN.

**Results**. Our results revealed that T2D patients have lower gut microbial diversity, no matter with or without SCH. The characteristic taxa of T2D patients were *Coriobacteriales*, *Coriobacteriaceae*, *Peptostreptococcaceae*, *Pseudomonadaceae*, *Collinsella*, *Pseudomonas* and *Romboutsia*. Meanwhile, *Escherichia/Shigella*, *Lactobacillus_Oris*, *Parabacteroides Distasonis_ATCC_8503*, *Acetanaerobacterium*, *Lactonifactor*, uncultured bacterium of *Acetanaerobacterium* were enriched in T2D patients with SCH. Moreover, serum levels of free triiodothyronine (FT3) and free thyroxine (FT4) in T2D patients were both negatively correlated with richness of gut microbiota. A number of specific taxa were also associated with clinical parameters at the phylum and genus level. In contrast, no correlation was found between AMY1 CN and T2D or T2D_SCH.

**Conclusion**. This study identified characteristic bacterial taxa in gut microbiota of T2D patients with or without SCH, as well as the taxa associated with clinical indices in T2D patients. These results might be exploited in the prevention, diagnosis and treatment of endocrine disorders in the future.

## INTRODUCTION

Diabetes and thyroid dysfunction are the two most common endocrine diseases, and they have become public health concerns in China due to their increasing prevalence, multiple complications and lifelong treatment (*Lambrinou, Hansen & Beulens, 2019*; *Shan et al., 2016*; *Wang et al., 2017*). Numerous studies have suggested a possible interaction between two diseases. On the one hand, both type 1 and type 2 diabetics have a higher incidence of thyroid diseases than general population, with subclinical hypothyroidism (SCH) occurs most frequently in diabetic patients (*Sotak, Felsoci & Lazurova, 2018*). On the other hand, patients with hyperthyroidism and hypothyroidism are also at increased risk for diabetes (*Biondi, Kahaly & Robertson, 2019*).

The causes of endocrine metabolic diseases are very diverse and complex, mainly including genetic predisposition and environmental factors (*Hu, 2003*). Environmental factors are relatively controllable and play a very important role: increasing studies have confirmed that environmental factors for example diet and physical activity, are closely related to obesity, diabetes, other endocrine and metabolic diseases (*Jebb & Moore, 1999*; *Lambrinou, Hansen & Beulens, 2019*). Gut microbiome has been neglected as symbiotic microorganisms. In the past, more attention has been paid to the association between endocrine metabolic diseases and common environmental factors such as lifestyles. However, they are not the whole story. Regarded as the ''virtual organ'' of human body, gut microbiota and their metabolites such as short-chain fatty acids (SCFAs) have been demonstrated to have great effects on host physiology, including energy homeostasis, vitamin synthesis, immune modulation and maintenance of gastrointestinal barrier (*LeBlanc et al., 2013*; *Natividad & Verdu, 2013*; *Sekirov et al., 2010*). A balanced gut microbiota requires a great number of microorganisms in the intestinal microecology to live in harmony by adopting mutualistic strategies, and the eubiosis condition of the gut microbiota is of great importance to human health. Thus, the alternation in the species and proportion of gut microbiota can affect the physiological condition of human body even promote the occurrence and development of certain gastrointestinal and systemic diseases including autoimmune diseases, metabolic disorders, mental diseases and even cancer (*Bhattarai, Muniz Pedrogo & Kashyap, 2017*; *Dinan & Cryan, 2017*; *Parekh, Balart & Johnson, 2015*; *Wong & Yu, 2019*).

Numerous studies have shown that gut microbes and SCFAs could indirectly regulate energy homeostasis and insulin signaling, as well as low-grade inflammation, which may further influence the pathogenesis of type 2 diabetes (T2D) (*Kimura et al., 2013*; *Qin et al., 2012*; *Tai, Wong & Wen, 2015*). *Zhang et al. (2013)* investigated the correlation between glucose intolerance and human gut microbiota in a Chinese population using 16S rRNA sequencing. It revealed the composition and diversity of gut microbiota had already changed in the prediabetes (*Zhang et al., 2013*). Moreover, a balanced gut microbiota is not only important in nutrients metabolism, immune regulation and maintenance of the gut barrier, but also beneficial for thyroid function. Increasing studies suggested that gut microbiota could modulate thyroid homeostasis through various mechanisms: on the one hand, it affects the synthesis of thyroid hormones by acting on the absorption

of thyroid-related micronutrients such as iodine, iron, copper and vitamin D (*Fröhlich & Wahl, 2019*; *Knezevic et al., 2020*); on the other hand, it influences the metabolism and storage of thyroid hormones by regulating immune responses, intestinal barrier and iodothyronine-deiodinases activity (*Nguyen et al., 1993*; *Sasso et al., 2004*; *Virili et al., 2018*). Recently, by using rodent model, *Khakisahneh et al. (2021)* demonstrated that both states of hyperthyroidism and hypothyroidism were associated with gut microbial diversity and composition. A cross-sectional study also revealed the correlation between L-thyroxine treatment and certain species of gut microbes in subclinical hypothyroidism subjects (*Yao et al., 2020*). However, the links between gut microbiota and thyroid function are still less systematically investigated.

Copy number (CN) variation (CNV) in human genome provides new insights into the heritability of human diseases including T2D, where CNV affects the disease susceptibility *via* shaping the gene expression level (*Bae et al., 2011*; *Santos et al., 2012*). The salivary $\alpha$-amylase gene (AMY1) encodes the enzyme responsible for dietary starch digestion, which typically ranges from one to 27 copies with a high within-population variability (*Fernandez & Wiley, 2017*). It is thought the CNV of AMY1 is the result of natural selection, as an increase in AMY1 CN could have been caused by the adaptation to high-starch foods during the Neolithic agriculture transition (*Kelley & Swanson, 2008*; *Perry et al., 2015*). Meanwhile, studies have reported that lower $\alpha$-amylase activity and AMY1 CN are associated with the prevalence of obesity, insulin resistance and the susceptibility to metabolic disorders (*Choi et al., 2015*; *Falchi et al., 2014*; *Nakajima et al., 2011*; *Viljakainen et al., 2015*). Indeed, higher $\alpha$-amylase activity and AMY1 CN are associated with faster digestion of starch food (*Atkinson et al., 2018*). In contrast, individuals with lower AMY1 CN are less able to digest starch (*Poole et al., 2019*). Given that the capacity of starch digestion is closely related to postprandial glucose level (*Atkinson et al., 2018*; *Mandel & Breslin, 2012*; *Nakajima, 2016*), which could further influence the incidence of insulin resistance and T2D, understanding the possible role of AMY1 genetic polymorphisms in T2D is of great importance, which could eventually benefit T2D patient with personalized nutrition.

Therefore, in order to clarify host-microbe interactions in T2D patients with thyroid disorders, and determine the potential role of genetic and environmental factors in glucose metabolism and thyroid homeostasis, here we investigated the profile of gut microbiota and AMY1 CNV in T2D patients with or without SCH by 16S rRNA sequencing and droplet digital PCR (ddPCR) separately, and identified specific taxa of gut microbiota associated with T2D and SCH. Additionally, the correlations between clinical parameters and gut microbiota in patients were also analyzed, and it was found that clinical indicators related to glucose metabolism and thyroid function also correlated with some specific taxa. This study may help enrich the understanding regarding to the profound association between human gut microbiota and endocrine diseases.

# MATERIALS & METHODS

## Participants

A total of sixty participants were enrolled between August and November, 2020, including 30 euthyroid T2D patients and 15 T2D patients with SCH recruited from the inpatient

department of endocrinology and 15 healthy individuals recruited from the medical examination center at Xinxiang First People's Hospital. Three groups were matched for gender, age and BMI. The diagnostic criteria of T2D were based on the 1999 WHO diagnostic criteria for diabetes, and SCH was diagnosed as TSH>4.2 mIU/L without any abnormality of FT3 and FT4. None of the patients had received thyroid hormone therapy. Subjects containing the following conditions were excluded: age<18 years or >70 years, a known history of severe trauma, surgery and bowel diseases, use of antibiotics or probiotics within last three months, pregnancy, lactation, suffer from acute or chronic diarrhea recently, and a known history of other endocrine diseases.

All procedures and amendments of this study were approved by the Ethics Committee of School of Public Health, Lanzhou University (IRB20033001). Informed consent was obtained from all participants enrolled in this study.

## Sample collection

Fecal samples were collected with screw-cap sterile plastic bottles. Once collected, the stool samples were transferred to the laboratory on dry ice and stored in $-80\,°$ C. After an overnight fast ($\geq$8 h), peripheral blood samples of the hospitalized patients (T2D group and TD_SCH group) were collected for further medical examination, including FBG, PBG, HbA1c, FT3, FT4, TSH, TG, total cholesterol TCH, HDL-C, LDL-C and serum albumin.

## Droplet digital PCR analysis of AMY1 CN

Human genomic DNA was isolated from stool samples following manufacturer's protocol (isolation of DNA from stool for human DNA analysis, QIAamp DNA Stool Mini Kit, QIAGEN). The copy number of the AMY1 gene was determined by ddPCR using QIAcuity One Digital PCR System (Qiagen, Hilden, Germany). The primers and probes used to detect target gene AMY1 and reference gene Near_AMY are (*Usher et al., 2015*): AMY1_assay2-forward: 5′-TGTTTGCAAGGAGGTCTTCTC-3′, AMY1_assay2-reverse: 5′-TTGGCCTTTCATCTGTGATTT-3′, and AMY1_assay2 Probe 5′-FAM-AAATGATTCCCGAAACTGTAGC-MGB-3′; Near_AMY-forward: 5′-AAATTTATTGGAGGGATGTTGG-3′, Near_AMY-reverse: 5′-TTCAAGTTTGACTGCT AACTCCTG-3′, and Near_AMY Probe 5′-VIC TGGAATAAAGAATCATTGGGCACAGGT-MGB-3′. The data were acquired using QuantStudio 3D Analysis Suite software. AMY1 CN was further calculated by the ratio between AMY1 and Near_AMY CN.

## 16S rRNA sequencing

Bacterial genomic DNA was isolated from stool samples following the manufacturer's protocol (isolation of DNA from stool for pathogen, QIAamp DNA Stool Mini Kit, QIAGEN). V3 and V4 regions of 16S rRNA was amplified with following primers: forward primer: 5′-CCTACGGGNGGCWGCAG-3′and reverse primer: 5′-GACTACHVGGGTATCTAATCC-3′, with adaptor sequences and sample-specific index sequences. The amplicons were sequenced with NovaSeq 6000 SP Reagent Kit (Illumina, San Diego, CA, USA) by Illumina NovaSeq 6000 sequencer (Illumina, San Diego, CA, USA).

## Bioinformatic analyses

The raw reads were processed in QIIME 2. The adaptor and primer were removed by Cutadapt plugin. Quality control and amplicon sequence variants (ASVs) exaction were conducted by DADA2 plugin. Taxonomic assignments of ASV representative sequences were conducted by Naive Bayes classifier trained on the Greengenes (version 13.8) with confidence threshold 0.8. Mothur was used to analyze the microbial alpha diversities among groups. Principal Coordinate Analysis (PCoA) based on Bray-Curtis distance matrices and partial least squares discriminant analysis (PLS-DA) were used to assess beta diversity among groups (*Lee, Liong & Jemain, 2018*). Partition Around Medoids (PAM) clustering over Jensen–Shannon divergence (JSD) distance was used to analyze enterotype composition. Specifically, JSD matrix was calculated based on the relative abundance in the community at genus level. The optimal number of clusters was determined by the Calinski-Harabasz (CH) index (*Arumugam et al., 2011*). Permutational multivariate analysis of variance (PERMANOVA) was conducted to assess the significance of the difference within gut microbiota composition based on the UniFrac distances with 9999 permutations (*Schnorr et al., 2014*). Linear discriminant analysis effect size (LEfSe) was performed in a web-based platform (*Afgan et al., 2018*) with a threshold of LDA score set at 2.0 to screen characteristically taxa that can most likely to explain the difference of gut microbiota among three groups.

## Statistical analysis

Kolmogorov–Smirnov test was used to test the normality of continuous variables. Chi-square test was used to analyze whether the distribution of sex was different among groups. The distribution of age, BMI, AMY1 CN and alpha diversity parameters of gut microbiota among three groups were compared by using Kruskal-Wallis test. The association between serum biochemical indices and gut microbial diversity and composition was analyzed by Spearman's correlation. All analysis was performed by using IBM SPSS software version 21 (IBM, Chicago, IL, USA). $P<0.05$ were considered significant.

# RESULTS

## Baseline characteristics of participants

Total of 60 subjects participated in our study, including 30 euthyroid T2D patients (T2D group), 15 T2D patients with SCH (T2D_SCH group) and 15 healthy controls (C group). Gender, age, and body mass index (BMI) of three groups were matched. As shown in Table 1, the distribution of them were not significantly different among three groups. Clinical characteristics including serum biochemical parameters and medication of T2D group and T2D_SCH group are listed in Table S1.

Figure 1A indicated the distribution of AMY1 CN of participants, which ranged from 1 to 14 with a median of 3.59. The AMY1 CN in T2D group ranged from one to nine, and ranged from one to 8 in T2D_SCH group. In control group, it was from 1 to 14. As shown in Fig. 1B, the median AMY1 CN among T2D group (4.29), T2D_SCH group (3.29) and C group (3.30) are not statistically significant. Meanwhile, no correlation was found between AMY1 CN and clinical parameters in patients with T2D and T2D_SCH (Fig. S1).

**Table 1  Basic characteristics of study participants.**

|  | T2D | T2D_SCH | Control | *P* value |
|---|---|---|---|---|
| *n* | 30 | 15 | 15 |  |
| Gender (male) | 20 (66.7%) | 8 (53.3%) | 8 (53.3%) | 0.573 |
| Age (years) | 53.97 ± 9.33 | 50.53 ± 9.93 | 49.6 ± 6.61 | 0.236 |
| BMI (kg/m²) | 25.03 ± 2.93 | 25.46 ± 2.82 | 24.58 ± 2.67 | 0.672 |

**Notes.**
Age and BMI data are shown as mean ± SD.
BMI, body mass index; T2D, type 2 diabetes; T2D_SCH, type 2 diabetes with subclinical hypothyroidism.

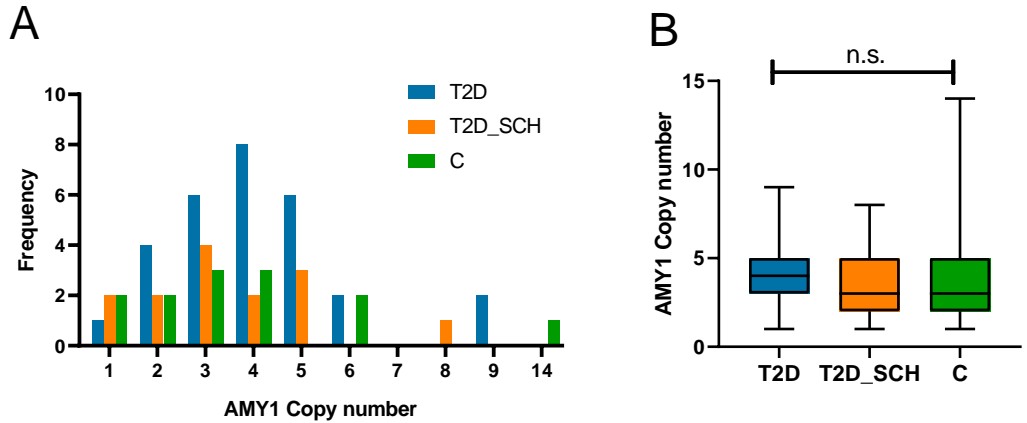

**Figure 1  The distribution of AMY1 CN in the study population.** (A) The distribution of AMY1 copy number in different groups. The AMY1 CN was rounded to nearest integer. (B) The median of AMY1 CN was compared among T2D, T2D_SCH and C group. n.s. indicates no significant difference ($P > 0.05$).

## Microbiome profile of the study population

A total of 5,674,655 raw reads were obtained from all 60 fecal samples *via* high-throughput sequencing. After carrying out a series of quality control and sequence optimization measures, 3,946,444 high-quality reads from all samples were collected with nearly 30% of raw data were filtered.

Overall gut microbiota composition of all subjects at the phylum level was shown in Fig. 2. The dominant bacterial phyla of all participants were Firmicutes (62.35%), Bacteroidetes (14.56%), Proteobacteria (13.10%), and Actinobacteria (8.88%). Both the distribution of four phyla ($P = 0.544$; 0.223; 0.531; 0.180) and the ratio of Firmicutes/Bacteroidetes ($P = 0.344$) is not statistically different among three groups.

Next, the gut microbial richness (ACE and Chao1 indices) and diversity (Shannon and Simpson indices) among three groups were compared. As shown in Figs. 3A and 3B, compared with healthy control, both T2D patients and T2D patients with SCH have lower gut microbial diversity, as indicated by significantly lower Shannon index and higher Simpson index ($P = 0.032$; 0.028). However, the difference between T2D_SCH group and T2D group was not statistically significant ($P = 0.608$; 0.706). As for gut microbial richness, ACE and Chao1 indices exhibited an increasing trend in healthy control, although

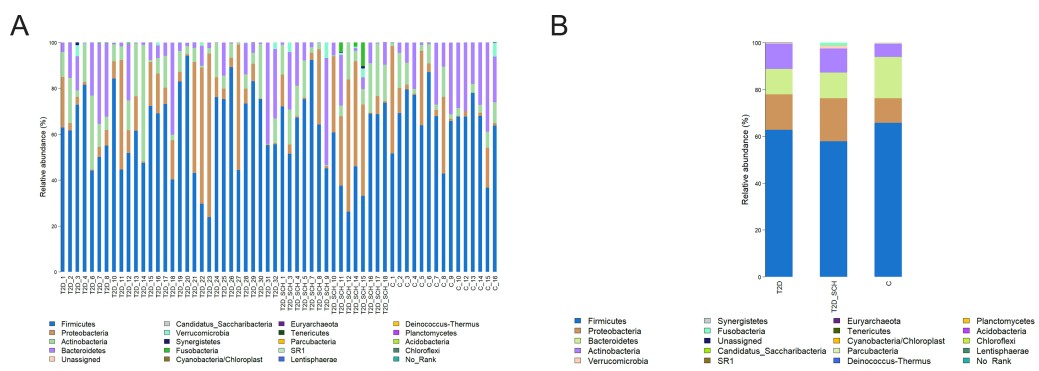

**Figure 2** Gut microbiota composition at the phylum level of all participants (A) and each group (B).

the difference was not statistically significant among three groups ($P = 0.223$; 0.208) (Figs. 3C and 3D). Low gut microbial diversity often occurs in individuals with other metabolic and inflammatory disorders such as obesity, non-alcoholic fatty liver disease, high blood pressure, and inflammatory bowel disease (*Le Chatelier et al., 2013*; *Shen et al., 2017*; *Sun et al., 2019*).

In terms of the similarity of microbiota communities among groups, a clustered tendency, which represented a relatively stable microbial community was shown in control group (Fig. 4A). The microbial beta diversity among three groups was significantly different ($P < 0.01$), despite that these groups were not distinctly separated on the PCoA plots. However, when PERMANOVA was conducted only between T2D and T2D_SCH groups, the beta diversity was not significantly different (Fig. S2). To further analyze the structural difference of gut microbial community among three groups, a supervised method-partial least squares discriminant analysis (PLS-DA) was performed across all samples. The result showed that the three groups were distinctly separated on the PLS-DA plots (Fig. 4B). Moreover, we further analyzed the microbiota according to the distribution of dominant genera by PAM-JSD method, and all the 60 samples were stratified into three enterotypes: enterotype 1 (*Ruminococcaceae*), enterotype 2 (*Prevotella*) and enterotype 3 (*Bacteroides*). It was confirmed that *Ruminococcaceae*, *Prevotella* and *Bacteroides* were predominant in enterotype 1, enterotype 2 and enterotype 3 (Fig. S3). Then PCoA analysis was performed at the genus level to visualize the three enterotypes, which showed that their microbial composition was distinctly different (Fig. 4C, $P<0.01$). The distribution of the three enterotypes among three groups was also analyzed, however, there was no statistical significance (Fig. 4D, Fisher's exact test $P = 0.083$).

## Microbiome signatures of T2D patients with or without SCH

To identify the characteristic bacterial taxa of T2D patients and T2D patients with SCH, linear discriminant analysis effect size (LEfSe) analysis was used to compare the composition of gut microbiota among three groups. As shown in Fig. 5, 18 discriminative features have been identified among three groups. The characteristic taxa of T2D group were *Coriobacteriales*, *Coriobacteriaceae*, *Peptostreptococcaceae*,

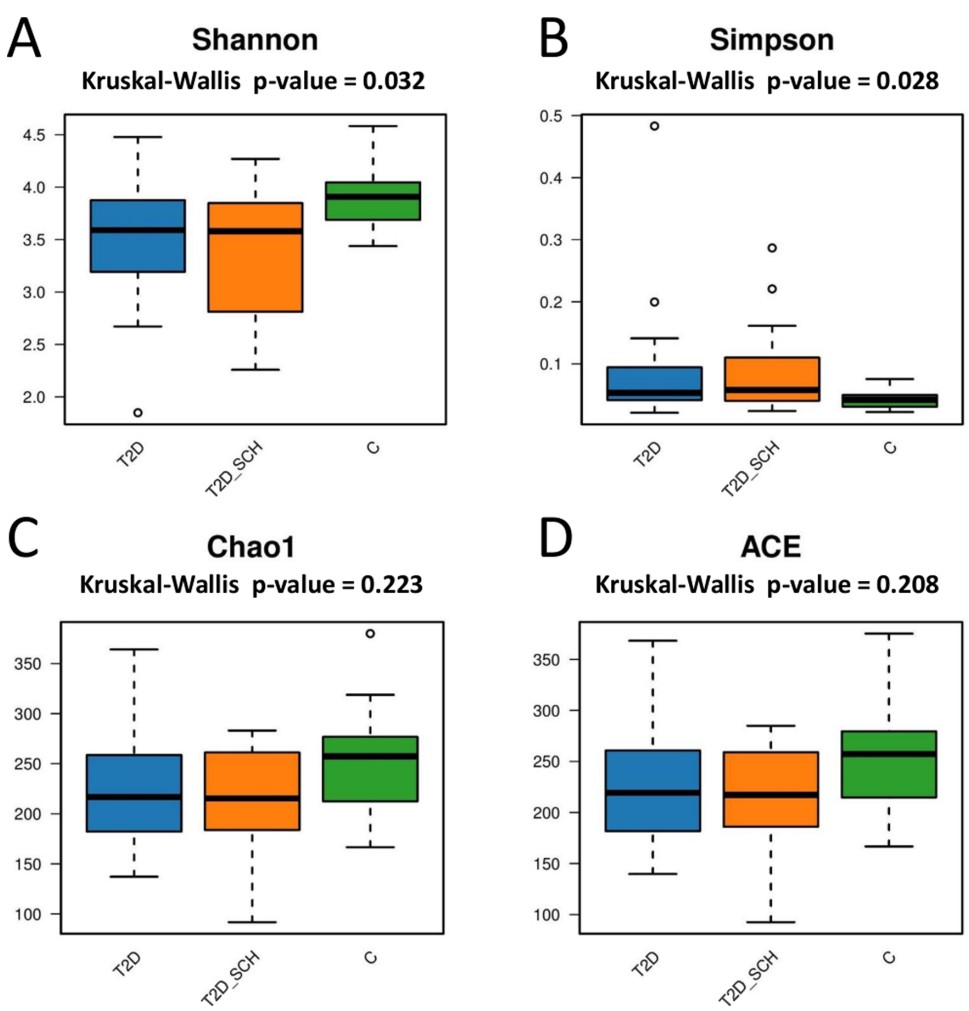

**Figure 3** Box plots of alpha diversity among three groups on the basis of Shannon (A) Simpson (B) ACE (C) and Chao1 (D) indices, all base on ASV level.

*Pseudomonadaceae, Collinsella, Pseudomonas* and *Romboutsia* (Fig. 5A). Among them, the genus *Collinsella* belongs to the *Coriobacteriaceae* family, *Coriobacteriales* order, the genus *Romboutsia* belongs to the *Peptostreptococcaceae* family and the genus *Pseudomonas* belongs to *Pseudomonadaceae* family (Fig. 5B). *Escherichia/Shigella, Lactobacillus_Oris, Parabacteroides_Distasonis_ATCC_8503, Acetanaerobacterium, Lactonifactor* and uncultured bacterium of *Acetanaerobacterium* were enriched in the T2D_SCH group. Among them, *Collinsella* was also reported enriching in T2D patients by Zhang et al. (*Zhang et al., 2013*), and *Escherichia-Shigella* was reported to involve in production of secondary bile acid, increase of which was observed in serum of overweight T2D patients (*Suhre et al., 2010*). In the control group, *Faecalibacterium, Coprococcus, Coprobacillus, Odoribacter* and *Bacteroides_intestinalis* were more concentrated. These taxa may be used as potential biomarkers for discrimination of T2D and T2D_SCH, which may help to distinguish T2D patients with SCH and to assess the risk of SCH in T2D patients.

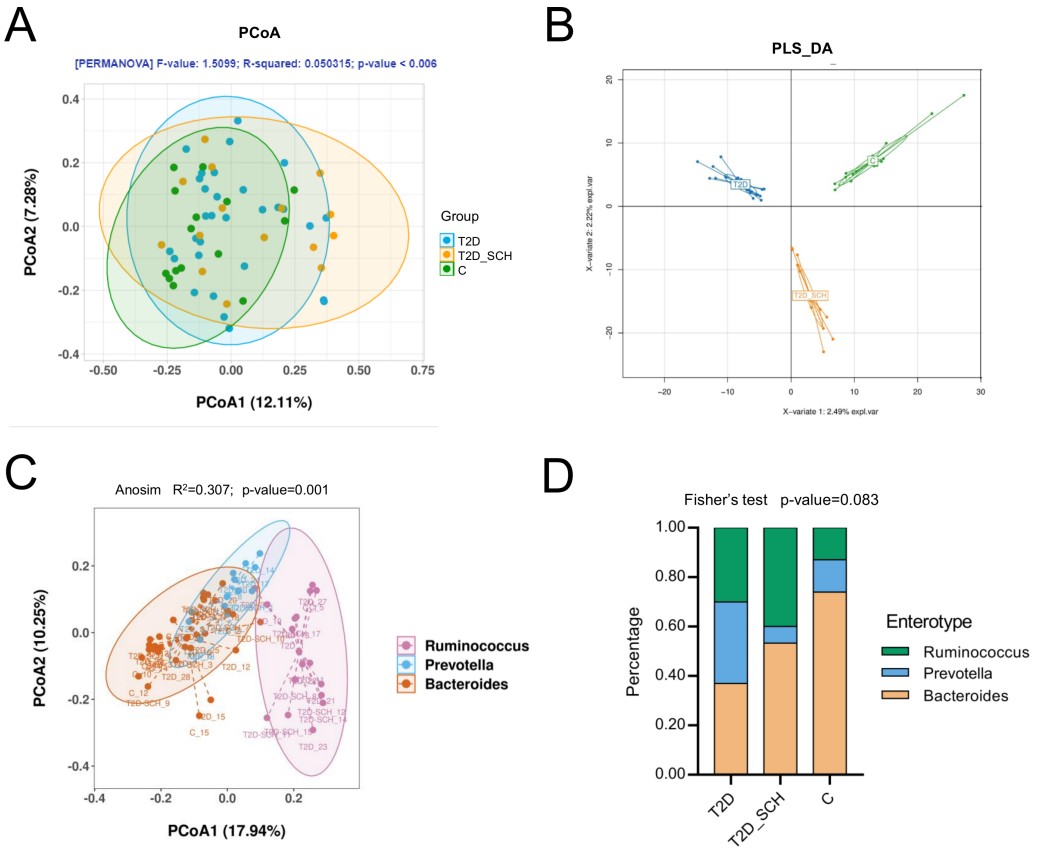

**Figure 4 Beta diversity of gut microbiota and gut enterotypes of all samples.** (A) Beta diversity described by PCoA based on Bray-Curtis distance matrices. (B) Beta diversity described by PLS-DA. (C) Enterotypes visualized by PCoA, in which 60 samples were clustered into three types at the genus level. (D) The distribution of three enterotypes in T2D, TD-SCH and C group. For PCoA analyses, the percent of variation explained by each axis was shown in axis titles. Each sample corresponded to one dot in the graphs. The circle summarized the area of gathering of the dots.

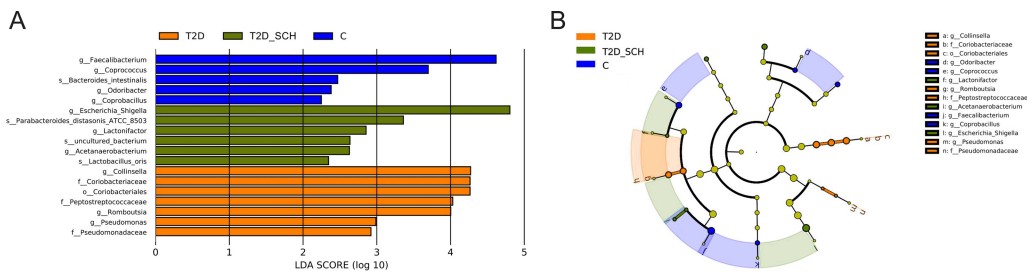

**Figure 5 Bacterial features most likely explain differences among groups identified by LEfSe based on ASV level.** (A) Histogram of the LDA scores. (B) LEfSe taxonomic cladogram. The letter in the former of the name of bacteria indicates different taxa levels (''g'' indicates genus; ''s'' indicates species; ''f'' indicates family).

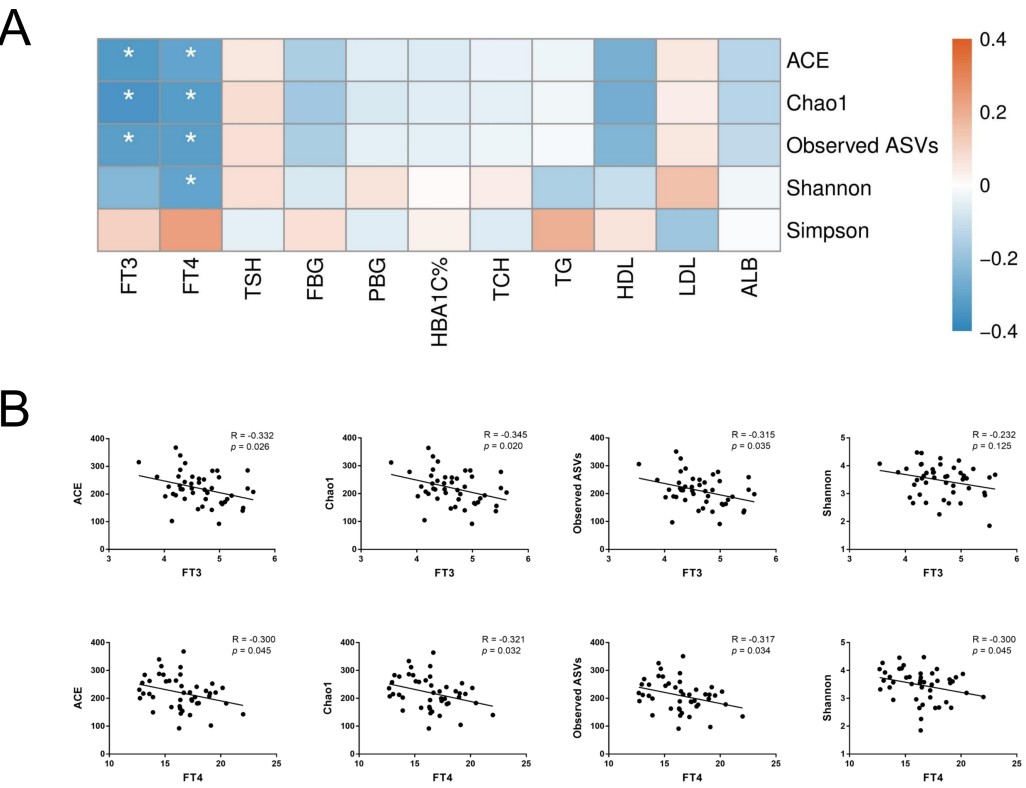

**Figure 6** **Correlation between host clinical parameters and gut microbial diversity.** (A) Heatmap of correlations between gut microbial diversity and clinical indicators of T2D patients. The color bar with numbers indicates the correlation coefficients. *P*-values are shown by asterisks (*: *P* < 0.05). (B) Correlation scatter plots of thyroid hormones and gut microbial diversity.

## Correlation analysis of clinical indices and gut microbiota

To further assess the relationship between gut microbiota and host metabolic status mainly including glucose metabolism and thyroid function, Spearman's rank correlation was conducted in T2D and T2D_SCH groups. First, the correlation between host clinical parameters and gut microbial diversity was explored. As shown in Fig. 6, serum levels of both free triiodothyronine (FT3) and free thyroxine (FT4) were negatively correlated with gut microbial richness (ACE and Chao1 indices). Moreover, FT4 level was also negatively correlated with Shannon index. These results demonstrated that gut microbial richness decreased as serum FT3 and FT4 levels increased, and increased FT4 level may also be accompanied with a low gut microbial diversity in T2D patients.

The correlation between abundances of gut microorganisms and clinical parameters of T2D patients was also analyzed. As shown in Fig. 7, at the phylum level, serum FT3 level was correlated with the relative abundance of Firmicutes and Bacteroidetes, indicating that with the increase of FT3, the proportion of Firmicutes increased and that of Bacteroidetes decreased. And there were negative correlations between fasting blood glucose (FBG) level and the abundances of both Parcubacteria and Proteobacteria, while postprandial blood glucose (PBG) was only negatively correlated with Parcubacteria. Serum lipid levels were

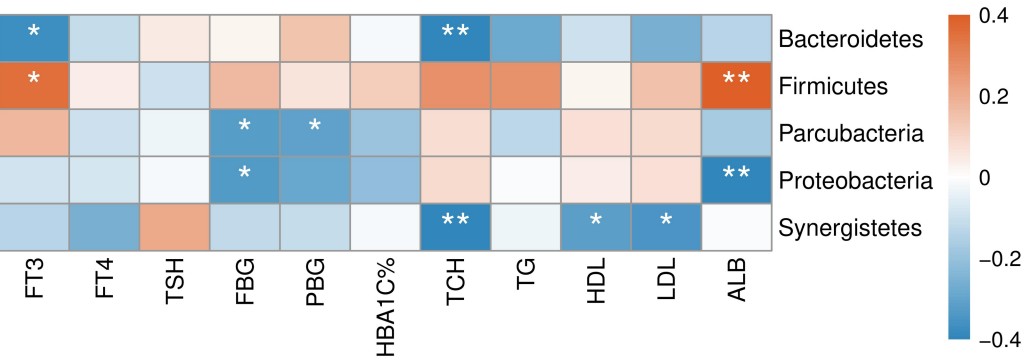

**Figure 7 Heatmap of correlations between abundances of intestinal bacteria and clinical indicators of T2D patients at the phylum level.** The color bar with numbers indicates the correlation coefficients. *P*-values are shown by asterisks (*: $P < 0.05$; **: $P < 0.01$).

also correlated with some specific taxa at the phylum level: total cholesterol (TCH) level was negatively correlated with both Bacteroidetes and Synergistetes, and both high- (HDL-C) and low-density lipoprotein cholesterols (LDL-C) levels were negatively correlated with Synergistetes. Moreover, serum albumin was correlated with the abundances of Firmicutes positively, and the abundances of Proteobacteria negatively. At present, the contribution of Firmicutes or Bacteroidetes in the incidence of T2D is still being debated. In our study though no correlation between blood glucose and Firmicutes or Bacteroidetes was observed, here we showed that the level of FT3 in T2D patients was correlated with the abundance of Firmicutes positively, and the abundance of Bacteroidetes negatively. Thus, the debate on the contribution of Firmicutes to T2D may consider more about other physiology perspectives such as thyroid function.

At the genus level (Fig. 8), clinical indicators related to glucose metabolism and thyroid function also correlated with some specific taxa: FBG level was correlated with *Parasutterella* positively, but *Escherichia/Shigella* negatively. There is evidence demonstrating that *Parasutterella* in gut could change the metabolism of aromatic amino acid, bile acid derivatives, etc. in mice, suggesting that *Parasutterella* is closely related to various metabolic processes (*Ju et al., 2019*). No taxon was correlated with PBG and hemoglobin A1c (HbA1c) levels. As for the association between thyroid function and abundances of gut microbiota, *Adlercreutzia, Blautia, Clostridium_XVIII, Fusicatenibacter, Gemella, Lachnospiracea_incertae_sedis* and *Rothia* were positively correlated with serum FT3 level, while *Alistipes, Bacteroides, Bilophila, Catenibacterium, Oscillibacter* and *Parabacteroides* were negatively correlated with FT3. FT4 level was to decrease in line with the increase of *Anaerobacterium, Anaerovorax, Clostridium_IV, Enterococcus* and *Slackia*. And thyroid stimulating hormone (TSH) level was positively correlated with *Coprobacillus, Lactobacillus, Parabacteroides* and *Pediococcus*, while negatively correlated with *Acinetobacter* and *Rcinetobacter*. In terms of serum lipid levels, both TCH and triglyceride (TG) levels were positively correlated with *Granulicatella, Gemella, Leuconostoc* and *Solobacterium*, while negatively correlated with *Anaerotruncus* and *Oscillibacter*.

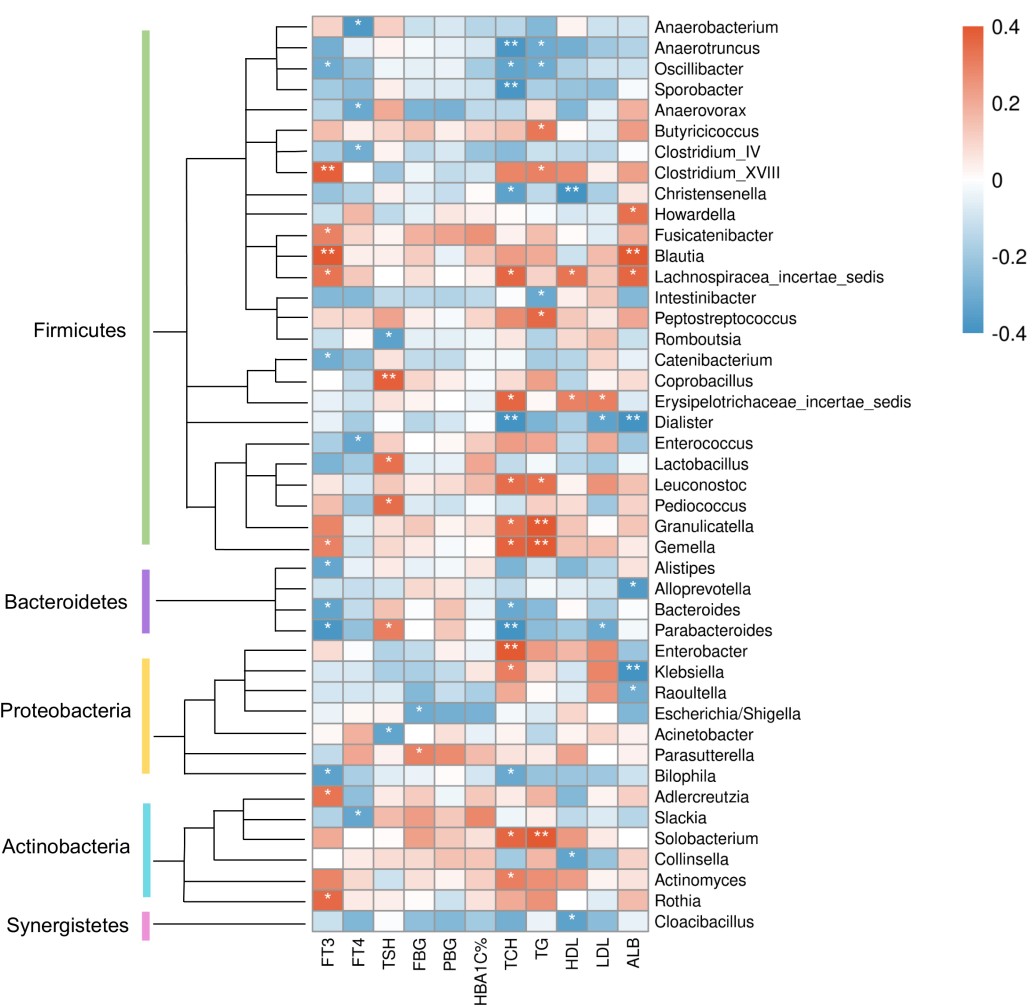

**Figure 8** Heatmap of correlations between abundances of intestinal bacteria and clinical indicators of T2D patients at the genus level. The color bar with numbers indicates the correlation coefficients. *P*-values are shown by asterisks (*: $P < 0.05$; **: $P < 0.01$).

*Erysipelotrichaceae_incertae_sedis* was to decrease in line with the increase of all TCH, HDL-C and LDL-C levels. Both TCH and LDL-C were negatively correlated with *Dialister* and *Parabacteroides*; both TCH and HDL-C were correlated with *Lachnospiracea_incertae_sedis* positively, but correlated with *Christensenella* negatively. Apart from the taxa above, TCH was also positively correlated with *Actinomyces*, *Enterobacter* and *Klebsiella*, while negatively correlated with *Bacteroides*, *Bilophila* and *Sporobacter*. TG was positively correlated with *Butyricicoccus*, *Clostridium_XVIII* and *Peptostreptococcus*, while negatively correlated with *Intestinibacter*. And HDL-C was to decrease in line with the increase of *Cloacibacillus* and *Collinsella*. Meanwhile, AMY1 CN was found to be correlated with specific taxa of gut microbes (Table S2). At the genus level, AMY1 CN were positively correlated with *Granulococcus*, *Klebsiella*, *Escherichia/Shigella* and *Rothia*, while negatively correlated with *Sporobacter*, *Anaerovorax*, *Clostridium_IV*, *Alistipes*, *Bacteroides* and *Parabacteroides*.

## DISCUSSION

In this study, the characteristics of gut microbiome and AMY1 CNV in T2D patients with or without SCH were explored. The correlation between the clinical parameters and gut microbiota composition and diversity was analyzed in patients as well.

Although it is hard to define exactly what a balanced gut microbiota is, dysbiosis of flora is often associated with excess of pathobionts and reduced microbial diversity (*Levy et al., 2017*). In our study, compared to healthy individuals, the gut microbial diversity of patients with T2D was significantly reduced, which was consistent with previous studies (*Chávez-Carbajal et al., 2020*; *Jandhyala et al., 2017*; *Leiva-Gea et al., 2018*). However, we failed to get statistically significant result in the reduction of gut microbial richness in T2D patients, similar result was reported in a Denmark study targeting on T2D patients (*Larsen et al., 2010*). We also found that T2D patients with SCH have lower gut microbial diversity compared to healthy individuals, although the difference between T2D_SCH group and T2D group was not statistically significant. Moreover, one of our key findings was that serum levels of both FT3 and FT4 were negatively correlated with gut microbial richness (ACE and Chao1 indices). And FT4 level was also negatively correlated with microbial diversity (Shannon index) in T2D patients. Similarly, another study targeting Chinese population showed that patients with primary hypothyroidism had significantly less diversity of gut microbiota than healthy controls, and transplantation of the patients' gut microbiota resulted in hypothyroidism in mice (*Su et al., 2020*). There is no doubt that high diversity of microbial taxa provides a more exhaustive crosstalk between the microbial and host metabolism (*Palau-Rodriguez et al., 2015*).

In our study, no significant difference in the composition of gut microbiota was found among three groups at the phylum level. Previous studies demonstrated that higher ratio of Firmicutes/Bacteroidetes was linked to Western lifestyles and obesity (*De Filippo et al., 2010*; *Ley et al., 2005*). However, recent studies indicated that the abundance of Firmicutes appeared to be more associated with impaired carbohydrate metabolism than obesity (*Muñoz Garach, Diaz-Perdigones & Tinahones, 2016*). At the same time, the contribution of Firmicutes or Bacteroidetes in the incidence of T2D is still being debated. One study found increased abundance of Firmicutes and reduced Bacteroidetes in obese T2D patients in which Firmicutes also appeared a positive correlation with the FBG level (*Ahmad et al., 2019*). In contrast, a significant reduction in the proportion of the Firmicutes phylum was observed in T2D patients in another study, where the ratio of Bacteroidetes to Firmicutes was positively correlated with blood glucose concentration, but not with BMI (*Larsen et al., 2010*), and even in prediabetes, the decreased abundance of Firmicutes was also observed (*Chávez-Carbajal et al., 2020*). Moreover, the debate on the contribution of Firmicutes to T2D may consider more about other physiology perspectives such as thyroid function. According to recent researches, the Firmicutes/Bacteroidetes ratio was significantly elevated in primary hypothyroidism patients (*Su et al., 2020*), and the same result was also reported in euthyroid patients with Hashimoto's thyroiditis (*Zhao et al., 2018*). It was also indicated that Firmicutes was enriched in patients with thyroid carcinoma (*Feng et al., 2019*). In fact, both Firmicutes and Bacteroidetes are involved in producing SCFAs, thus more

concentration should be paid on specific species rather than just comparing at the phylum level (*Cani et al., 2012*). Additionally, in this study, enterotype analysis was also performed and all samples were stratified into three enterotypes. Although no significant difference in enterotype distribution has been found among three groups, there is a tendency that the proportion of enterotype 1 (*Ruminococcus*) was higher in T2D (30%) and T2D_SCH (40%) groups compared to healthy control (13.3%). Similar result has been reported by Al Bataineh et al., where *Ruminococcus* enterotype was enriched in T2D patients (*Al Bataineh et al., 2020*). Meanwhile, there is evidence that increased *Prevotella* abundance has been linked to T2D, which was also observed in our study (T2D *vs* C) (*Larsen et al., 2010*).

Characteristic bacterial taxa of T2D patients with or without SCH were found in our study, which may help to distinguish T2D patients with SCH and to assess the risk of SCH in T2D patients. Compare to T2D patients, T2D_SCH patients are more likely to suffer from diabetes complications such as diabetic nephropathy (*Han et al., 2015*; *Mansournia et al., 2017*). Moreover, SCH was suggested to be associated with higher risk of cardiovascular events and metabolic syndrome (*Delitala et al., 2017*; *Delitala et al., 2019*). Our results suggested some biomarkers associated with SCH in T2D patients, which were absent in patients with T2D only. However, due to the relatively small sample size in this study, our finding need to be verified in well-powered human studies of large population, since to our knowledge, our study represents the first investigation on the characteristic taxa of T2D patients with SCH. Meanwhile, despite that very small differences may not be found due to the small sample size, we did find certain significant different taxa among three groups, and some are in consistence with previous reports.

Correlation analysis in this study also demonstrated that FBG level of T2D patients was positively correlated with *Parasutterella*, while negatively correlated with *Escherichia/Shigella*. Interestingly, in our previous study, *Parasutterella* was also found to be a characteristic taxon of overweight/obese people (BMI $\geq$ 24 kg/m$^2$) (*Lv et al., 2019*). Meanwhile, *Jang et al. (2019)* showed that the abundance of *Parasutterella* is significantly reduced in bodybuilders. Also, study in rats have demonstrated that the abundance of *Escherichia coli* (*E.coli*) were greater in T2D group than healthy controls (*Li, Zhang & Wang, 2020*). In contrast, lower abundance of *Escherichia/Shigella* in obese individuals with T2D was observed in a human study (*Ahmad et al., 2019*). Our study also demonstrated that *Collinsella* was one of the characteristic bacteria in T2D group. Consistent with this result, *Zhang et al. (2013)* found *Collinsella* had higher relative abundance in newly diagnosed T2D patients in a Chinese population. And higher abundance of *Collinsella* was also linked to other metabolic disorders such as obesity, atherosclerosis and nonalcoholic fatty liver disease (*Astbury et al., 2020*; *Gomez-Arango et al., 2018*; *Karlsson et al., 2012*).

Although less work has been done on the reciprocal influence between gut microbiota and thyroid homeostasis, intestinal flora was reported to play an important part in thyroid hormones metabolism and enterohepatic recycling (*Virili & Centanni, 2015*; *Virili & Centanni, 2017*). To our surprise, both probiotics (*Lactobacillus_Oris*) and opportunistic pathogens (*Escherichia/Shigella*) were defined as characteristic intestinal microbes in T2D patients with SCH. One study characterized the gut microbiota in hypothyroid patients with Hashimoto's thyroiditis, and found that compared to healthy subjects, the
abundance of *Escherichia/Shigella* increased in hypothyroid patients (*Ishaq et al., 2017*). Greater abundance of *E.coli* have also been observed in overweight pregnant women compared with normal weight ones (*Santacruz et al., 2010*). On the other hand, it has been demonstrated that certain components/proteins from specific strains of *Lactobacillus* and *Bifidobacterium* may act as antigenic properties of human autoantibodies by sharing amino acidic sequences with thyroid peroxidase and thyroglobulin (*Kiseleva et al., 2011*). According to one recent research, lower relative abundance of *Lactobacillus* was observed in thyroid cancer and thyroid nodules, and FT3 was negatively correlated with the genera *Lactobacillus* in thyroid nodules patients (*Zhang et al., 2019*). Zhou et al. also observed that both *Bifidobacterium* and *Lactobacillus* decreased in hyperthyroid individuals (*Zhou et al., 2014*). And some animal experiments showed that *Lactobacillus* has beneficial effect on thyroid gland homeostasis and bowel permeability (*Mu et al., 2017*; *Varian et al., 2014*). In term of patients with T2D, several studies showed the level of *Lactobacillus* was significantly higher in patients, while when it comes to *Bifidobacterium*, another widely recognized probiotic, the opposite result occurred (*Sato et al., 2014*; *Sedighi et al., 2017*; *Wu et al., 2010*).

Previous studies linking gut microbiome to endocrine diseases such as diabetes and thyroid disorders had different results. Complex influencing factors of gut microbiota make the reproducibility of studies reporting associations between the intestinal flora and diseases a challenge. The characteristics of participants (age, ethnicity, customs, etc.), differences in study methods and data analysis, may lead to the variations in results from different studies. Though these issues exist, research on the distribution of intestinal flora and its relationship with endocrine diseases is necessary in exploring the occurrence and development of several diseases, helping to better understand the underlying mechanism, and may benefit to disease prevention and treatment in the future.

Although the distribution and the median of AMY1 CN of our participants were comparable to what have been reported previously (*Falchi et al., 2014*; *Hasegawa et al., 2022*; *Marquina et al., 2019*), no significant correlation has been found between AMY1 CN and T2D or T2D_SCH. Indeed, some studies have described a negative association between AMY1 CN and diabetes in particular populations. For example, in a Korean population, AMY1 CN was found negatively correlates with insulin resistance and the incidence of T2D (*Choi et al., 2015*; *Shin & Lee, 2021*). In contrast, in Qatari women, the risk of diabetes was only associated with low salivary $\alpha$-amylase activity, but not low AMY1 CN (*Al-Akl, Thompson & Arredouani, 2021*). Similarly, another study reported a negative correlation between salivary $\alpha$-amylase activity, but not AMY1 CN, with fasting plasma glucose levels in a French population (*Bonnefond et al., 2017*). Studies investigating the correlation between obesity and AMY1 CN also failed to draw a consistent conclusion (*Falchi et al., 2014*; *Usher et al., 2015*; *Yong et al., 2016*). Besides the variations on study populations and methods, another possible explanation could be that AMY1 CN is not the solo factor determining the level of salivary $\alpha$-amylase (*Perry et al., 2007*). Other genetic and environmental factors such as oral health and diet also influence salivary $\alpha$-amylase production and activity (*Carpenter, Mitchell & Armour, 2017*; *Granger et al., 2007*; *Hasegawa et al., 2022*; *Heianza et al., 2020*; *Lawrence, 2002*; *Nater, Hoppmann & Scott, 2013*). Therefore, more attention

should be paid to the variation of enzyme itself, which together with AMY1 CNV could lead to a more convincing conclusion in studies targeting metabolic disorders.

However, some limitations still existed in the present study. We did not include SCH patients without T2D in our participants, so further investigations involving patients with SCH only are needed to better explore the association between endocrine metabolic diseases and gut microbiota. Since we may not have had sufficient information about patients' previous medication such as glucagon-like peptide-1 receptor agonists and statins, which may influence the gut microbiota composition (*Everard & Cani, 2014*; *Vieira-Silva et al., 2020*), further multivariate analyses are required to explore the contribution of medication to the gut microbiota of patients with endocrine diseases. Moreover, this is a case-control study that may fails to prove the chronological order of causes and consequences, thus, future prospective study and randomized controlled trial are warranted to further evaluate the causality. Meanwhile, due to the fact that the sample size of this study was limited, well-powered human studies of large population should also be carried out to further confirm the impact of alterations in gut microbiota on endocrine and metabolic diseases.

## CONCLUSIONS

In this study, characteristic bacterial taxa in gut microbiota of T2D patients with or without SCH were identified, which might be used as biomarkers in discriminating T2D patient with SCH and for clinical treatment in future. This work also provides insight into the relationship between clinical indices and gut microbiota in T2D patients, so as to find the possible association between human gut microbiota and both glucose metabolism and thyroid homeostasis, and eventually be exploited in the prevention, diagnosis and treatment of endocrine disorders.

## ACKNOWLEDGEMENTS

We are grateful to all the participants in this study.

### Funding

This research was supported by the National Natural Science Foundation of China (No. 81501734, 81803308), the Science and Technology Planning Project of Chengguan District in Lanzhou (No. 2021-9-12), and the Undergraduate Research Experience Program (No. 202210730201) from the China National Research Fund and Supercomputing Center of Lanzhou University. The funders had no role in study design, data collection and analysis, decision to publish, or preparation of the manuscript.

### Grant Disclosures

The following grant information was disclosed by the authors:
The National Natural Science Foundation of China: 81501734, 81803308.

The Science and Technology Planning Project of Chengguan District in Lanzhou: 2021-9-12.
The Undergraduate Research Experience Program from the China National Research Fund: 202210730201.
Supercomputing Center of Lanzhou University.

## Competing Interests

The authors declare there are no competing interests.

## Author Contributions

- Yanrong Lv and Rong Liu performed the experiments, analyzed the data, prepared figures and/or tables, authored or reviewed drafts of the article, and approved the final draft.
- Huaijie Jia performed the experiments, prepared figures and/or tables, and approved the final draft.
- Xiaolan Sun and Li Ma analyzed the data, authored or reviewed drafts of the article, and approved the final draft.
- Yuhan Gong analyzed the data, prepared figures and/or tables, authored or reviewed drafts of the article, and approved the final draft.
- Wei Qiu and Xiaoxia Wang conceived and designed the experiments, authored or reviewed drafts of the article, and approved the final draft.

## Human Ethics

The following information was supplied relating to ethical approvals (i.e., approving body and any reference numbers):

All procedures and amendments of this study were approved by the Ethics Committee of School of Public Health, Lanzhou University (IRB20033001). Informed consent was obtained from all subjects involved in the study.

## Data Availability

The datasets analyzed during the current study are available in the SRA database: ID PRJNA787412.

## Supplemental Information

Supplemental information for this article can be found online at http://dx.doi.org/10.7717/peerj.15193#supplemental-information.

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
