# Peer review of "Alterations of the gut microbiota in type 2 diabetics with or without subclinical hypothyroidism"

_PeerJ, doi:10.7717/peerj.15193_

## Round 0.1 · original submission · Major Revisions

While all 3 reviewers were positive, some issues have been highlighted which require your attention. Reviewers 1 and 3 are straightforward and should not take you much time.

However, reviewer-2 has made many good points which if properly addressed will add greatly to this work. Hence I urge you to address all of these, make corresponding changes to the paper and explain your changes rebuttal.

I look forward to seeing the revisions to this worthwhile and interesting study.

Reviewer 1 ·

Basic reporting

Copy number of salivary amylase, blood levels of relevant markers for DM and SCH and microbiome profile in a 30 patients with T2DM, 15 patients with T2DM and 15 healthy controls are compared. Sufficient background information including references are given. English is professional with some linguistic errors. The structure of the manuscript is appropriate and clear hypotheses addressed. Comments: The used methodology is sufficiently described but in the patient collective additional information (medication oral/insulin, distribution of the relevant parameters for the correlations) should be added.

Experimental design

The research is within the scope of the journal and has relevance. There are no ethical concerns. Methodology is clearly described. Comments: It would have been good to include also a group with SCH only for better identification of the influence of the disease entities.

Validity of the findings

Conclusion and discussion are extensive but linked to the original research question. Comments: As stated already by the authors, the low number of participants limits the power of the statistic analysis (power is not given). Further, a collective of patients with SCH without T2DM would have been good. This is a limitation, which should be mentioned but the respective data cannot be included since this would require a new ethic approval.

Additional comments

L.231: Fig. 5 should read Fig. 6

Reviewer 2 ·

Basic reporting

Increasing evidence shows the crucial role of gut microbiota in metabolic disorder such as T2D and understanding the microbial difference between T2D at different levels are important for the development of related therapy. Lv, Liu et al. described a human study in which they recruited T2D cohorts with and without subclinical hypothyroidism. In this manuscript, they reported the profile of gut microbiota in patient cohorts and its association with different clinical index, as well as CNV analysis of AMY1 gene. Overall, the manuscript is relatively well-written and well-organized, although not such significant biological conclusions were drawn in the manuscript, the data generated in this work as well preliminary analysis of microbial association would be beneficial to the microbiota&T2D field, thus I will support the publication of this work if the following issues could be addressed by the authors.

Data accession:

Although no significant conclusion was made, this study provides an important resource of human cohort study and 16S sequencing result, thus would be important to deposit the original data to public repository before publication, such as NCBI SRA, so other researchers could access it for other further comparative analysis.

Background:

Line-61: In my understanding, gut microbiota is not an environmental factor (such as diet and physical activity). However, compared to the human body, gut microbiota is an external factor rather than internal factors such as hormones. Would be better to rephrase the sentence.

Line-88, from the data generated in Khakisahneh et al. It's an overstatement to say the states could change the gut microbiota since the paper didn’t show causality analysis. Better to say “show difference” or “association” rather than “could change”

Line-97, this is a cool study, but the authors need to give more background about the following point to make the point clear (1) how CNV occur (2) frequency and ranges of CN in healthy and patients (3) association of CNV and other diseases

Line-104, use “profile” instead of “characteristics”

Line-106, use “specific” not “special”

Line-108, the correlation analysis is cool but the end of the paragraph is too sudden. Better to add more summary of result and significance here


Method:

Line-131, the authors mentioned that human genomic DNA extraction from feces thus the yield could be pretty low since majority of the human fecal DNA should be microbial. Please point out the yield of human genomic DNA (as well as the estimation of human % by qPCR) to make sure there is no molecular bottleneck and validate the fidelity of the data. For example, 0.1ng of 100% human DNA will be just 30 copies.

Result:

Line-182 and Figure-1: here I have an important concern: the CN analysis of the cohort is pretty clear but the result is not significant and the analysis of AMY1 CN seems pretty separated from the following microbiota as well as the whole manuscript. However, I do think the AMY1 CN data is interesting, but the authors definitely need to better re-analyze it or illustrate it. I would suggest as following: (1) the authors could try to link the AMY1 CN with the clinical index as well as some co-association analysis with gut microbiota to see if there are any interesting correlations as the non-significant result in Figure-1 is trivial; (2) if any significant correlations are identified, I would suggest making an illustration/model plot to elucidate potential links of association or potential underlying mechanism.

Figure-2: the authors showed phylum-level microbial abundance for each individual, but the aesthetics of the figure make it difficult to understand the figure. For example, the authors could change the colors of the bar to make it less glaring and put the averaged value of abundance for each group instead of each individual and make other panels for each group separately or point out a few examples to show variations.

Figure-3, please point out for which comparison the P-value was calculated.

Figure-4A, although the difference between groups is not obvious on the plot, the statistics shows significance. Here I would like to ask the author to incorporate the enterotypes of the cohorts in this analysis and maybe that would help to stratify and understand the heterogenicity and difference between groups.

Figure-5, this analysis is awesome and I would like to see more about the result, such like the association at the phylogenic tree to see if any phylogeny are linked. This will help to figure out potential metabolites that play the role.

Line-232, typo, Figure-5 but Figure-6

Line-234 and Figure-6, this correlation is an important data and maybe good to have a separate scatter plot to show the individual correlation between these indices.

Line-239, since the authors were measuring the relative abundance, and Firmicutes & Bacteroidetes are dominating phylum in gut microbiota, thus if one shows positive correlation, another should show negative. The authors may need to better phase the sentence.

Figure-8. The association analysis at genus levels is informative. I would like to ask the authors to reorder the genus (instead of alphabet order) according to the phylogeny to show the pattern and plot the phylogenetic tree on the side of the heatmap as reference.

Figure-7 & Figure-8, the authors mentioned the association analysis was performed on T2D patients and I would suggest performing the same analysis for healthy control cohort as well as T2D with and without SCH.

Discussion

The authors put lot of interpretation of result part in the discussion and I would suggest moving these interpretation and related studies to the relevant result section to make the story more logical and easier to understand.

Acknowledgement

Please add the relevant funding resource.

Experimental design

no comment

Validity of the findings

no comment

·

Basic reporting

Line 28. The authors used “T2D” abbreviation first. It is should be spelled out
Lines 67 – 68. The meaning of the definitions «fine balance» of gut microbiota and «alterations» in gut microbiota should be clarified
Line 81. The authors should not use term intestinal «flora». It would be better to follow to the current term "microbiota"
Lines 34-38, 219-224. All names of bacteriological taxa should be written in a single way (in italics)

Experimental design

no comment

Validity of the findings

no comment

---

## Round 0.2 · Minor Revisions

Could you please address the issue raised about colour schemes, then I can accept the paper. Thanks for taking the time to address all the comments in the previous round of revisions.

Reviewer 2 ·

Basic reporting

Appreciation to all the authors for making the additional improvements and analysis. I am very happy to see my concerns/suggestions have been fully addressed. I would recommend this manuscript for publication at PeerJ with one minor aesthetic suggestion:

For Figure-4C, please use a different color schemes from Figure-4A as the same color scheme in Figure-4A and C is a little confusing.

Experimental design

no comment

Validity of the findings

no comment

Additional comments

no comment

---

## Round 0.3 · accepted · Accept

Thanks for attending to this. Congratulations on an interesting study.